# OBJVANISH: PHYSICALLY REALIZABLE TEXT-TO-3D ADV. GENERATION OF LIDAR-INVISIBLE OBJECTS

## ABSTRACT

LiDAR-based 3D object detectors are fundamental to autonomous driving, where failing to detect objects poses severe safety risks. Developing effective 3D adversarial attacks is essential for thoroughly testing these detection systems and exposing their vulnerabilities before real-world deployment. However, existing adversarial attacks that add optimized perturbations to 3D points have two critical limitations: they rarely cause complete object disappearance and prove difficult to implement in physical environments. We introduce the text-to-3D adversarial generation method, a novel approach enabling physically realizable attacks that can generate 3D models of objects truly invisible to LiDAR detectors and be easily realized in the real world. Specifically, we present the first empirical study that systematically investigates the factors influencing detection vulnerability by manipulating the topology, connectivity, and intensity of individual pedestrian 3D models and combining pedestrians with multiple objects within the CARLA simulation environment. Building on the insights, we propose the physically-informed text-to-3D adversarial generation (Phy3DAdvGen) that systematically optimizes text prompts by iteratively refining verbs, objects, and poses to produce LiDAR-invisible pedestrians. To ensure physical realizability, we construct a comprehensive object pool containing 13 3D models of real objects and constrain Phy3DAdvGen to generate 3D objects based on combinations of objects in this set. Extensive experiments demonstrate that our approach can generate 3D pedestrians that evade six state-of-the-art (SOTA) LiDAR 3D detectors in both CARLA simulation and physical environments, thereby highlighting vulnerabilities in safety-critical applications.

## 1 INTRODUCTION

LiDAR-based 3D object detection systems are foundational to modern autonomous driving and robotics. Their ability to accurately perceive and localize surrounding objects plays a vital role in ensuring navigation safety, obstacle avoidance, and informed decision-making (Lang et al., 2019; Shi et al., 2019; 2020; Hu et al., 2022; Zhang et al., 2022; 2023; 2024). However, the failure of such systems to detect certain objects—especially vulnerable road users like pedestrians—can result in catastrophic outcomes. As these systems are increasingly deployed in safety-critical applications, rigorously evaluating their robustness is essential for ensuring real-world reliability.

To this end, adversarial attacks have emerged as an effective tool for probing the weaknesses of perception systems before they are exploited in practice. Existing LiDAR-based attacks primarily fall into three categories: sensor-level spoofing with external hardware (Cao et al., 2019a; Zhu et al., 2021), digital perturbations to point clouds (Hamdi et al., 2020), and mesh-based object insertions (Xiao et al., 2019; Cao et al., 2019b). While these approaches can degrade model performance under controlled settings, they suffer from two key limitations: ❶ they rarely achieve complete object disappearance, and ❷ they are often difficult to implement in real-world physical environments. More importantly, these methods operate within or near the domain of captured sensor data, and do not explore the adversarial potential of fully generated 3D content.

In this work, we explore a fundamentally different and underexplored direction: leveraging text-to-3D generative models to create physically grounded, semantically plausible 3D objects that can evade LiDAR-based perception. As such generative models continue to improve in controllability and photorealism, they pose a new threat: attackers can manipulate prompt semantics—without touching

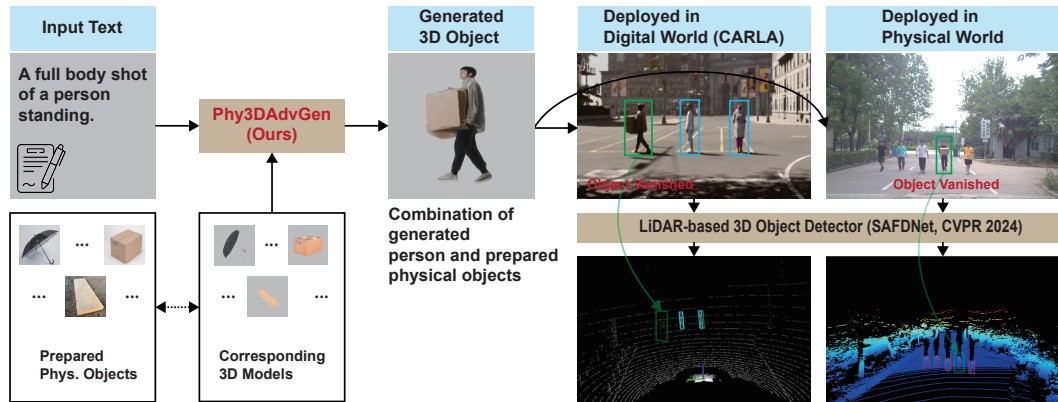

Figure 1: Example of Phy3DAdvGen for digital and physical deployment. Our method generates physically realizable adversarial 3D objects from discrete text prompts that successfully evade LiDAR detection (achieving "Object Vanished" status). To enable physical realization, we construct a prepared object pool and input the corresponding 3D models into our method. Phy3DAdvGen combines these prepared objects to generate adversarial 3D objects that are physically realizable without requiring 3D printing to obtain the optimized object in real-world scenarios.

sensor data—to generate adversarial 3D content that is both realistic and effective. Compared to point or mesh-level attacks, this paradigm allows greater variation and stronger attack potential. *This prompts a critical research question:* Can we adversarially optimize textual prompts to generate physically realizable 3D objects that fool SOTA LiDAR detectors in both simulation and reality?

To answer this question, we conduct the first systematic study of the factors affecting the detectability of 3D pedestrians in simulated LiDAR environments. By manipulating individual pedestrian attributes (*e.g.*, topology, connectivity, and intensity), combinations of pedestrians with multiple objects, and scene-level factors (*e.g.*, distance, angle, and vehicle speed), we assess their impact on detector performance within the CARLA simulation environment (Dosovitskiy et al., 2017). *Our study finds that combinations of multiple objects have the greatest impact on detection success, motivating the development of a prompt-based adversarial design that focuses on the semantics of object combinations.*

Building on this insight, we propose Phy3DAdvGen, a physically realizable text-to-3D adversarial generation framework. Our method constructs and optimizes discrete prompts composed of verbs, objects, and poses to generate 3D human-object compositions that are challenging for LiDAR detectors to perceive. These prompts are fed into a 3D generation backbone based on Gaussian Splatting (*e.g.*, LGM (Tang et al., 2024)), producing point-based representations that can be differentiably rendered into point cloud scenes. As illustrated in Fig. 1, Phy3DAdvGen generates human-object compositions that are visually realistic yet remain undetected in both simulated and real-world scenarios. To support real-world deployment, we constrain generated content to a curated pool of physical objects and use multi-view renderings to guide pose reproduction. This enables faithful physical realization of adversarial instances, which are then validated under actual LiDAR sensors—demonstrating that our attacks remain effective beyond simulation. Extensive experiments show that Phy3DAdvGen can consistently fool mainstream detectors (*e.g.*, (Zhang et al., 2024), HEDNet (Zhang et al., 2023)), PointRCNN (Shi et al., 2019)) across both digital and physical settings, revealing a practical and previously overlooked vulnerability in LiDAR 3D detectors.

Our contributions are summarized as follows:

- We present the first systematic study on the detectability of 3D objects in simulated LiDAR scenes. By manipulating 3D models, we assess how individual object attributes (*e.g.*, topology, connectivity, intensity), combinations of multiple objects, and scene-level conditions (*e.g.*, object distance, angle, vehicle speed) influence LiDAR detection performance, with object combinations emerging as the most influential factor.

- We propose Phy3DAdvGen, a novel framework that adversarially optimizes discrete textual components—verbs, objects, and poses—to generate realistic, LiDAR-invisible human-object compositions via a Gaussian Splatting-based generation pipeline.

- To ensure real-world feasibility, we constrain generation to a curated pool of physical objects and use multi-view rendering to supervise physical pose replication, enabling accurate physical reproduction of digital adversarial instances.

- We demonstrate that Phy3DAdvGen consistently fools state-of-the-art LiDAR detectors in both simulation and physical evaluations, revealing a practical and previously underexplored vulnerability in 3D perception systems.

## 2 RELATED WORK

### 2.1 ADVERSARIAL ATTACKS ON LIDAR-BASED DETECTORS.

**Spoofing attacks.** These sensor-level attacks manipulate LiDAR outputs through external interference—such as projected lasers or reflected signals—without accessing the internal pipeline. They are typically categorized as injection attacks, which add fake points to mislead detectors (Cao et al., 2019a; Shin et al., 2017; Jin et al., 2023; Sun et al., 2020; Wang et al., 2023), and removal attacks (Cao et al., 2023; Sato et al., 2024), which eliminate real points to cause missed detections.

**Point cloud perturbation attacks.** Another line of research directly perturbs LiDAR point coordinates to perform adversarial attacks. For instance, Wang et al. (2021) optimize a loss based on Intersection over Union (IoU), combined with imperceptibility constraints, to degrade the detection of car instances. To address detector non-differentiability, Liu et al. (2023a) design a surrogate loss that balances minimal point changes with maximal disruption. ShapeAdv Lee et al. (2020) introduces shape-aware adversarial attacks by perturbing the latent space of a point cloud auto-encoder to induce realistic geometric deformations that remain close to the original shape.

**Adversarial object attacks.** A third line of research focuses on inserting physically realizable 3D objects into LiDAR scenes to perform adversarial attacks. A common strategy is to directly optimize mesh geometry or texture, often using differentiable LiDAR rendering (Möller & Trumbore, 2005) to enable gradient-based shape optimization. For example, Cao et al. (2019b) employ differentiable ray casting to generate 3D-printable shapes that evade detection in real-world scenarios. Other works (Tu et al., 2020; Yang et al., 2021; Zheng et al., 2025; Xiao et al., 2019) insert predefined structures and refine their shape or appearance to interfere with nearby object detection.

Spoofing attacks require specialized equipment and expertise, while point perturbation attacks rely on acquired sensor data, making both challenging in physical realizability and deployment. In contrast, adversarial object attacks allow easy deployment but often depend on manually designed or optimized non-semantic meshes with limited scalability, optimization, and stealth. Our work enables large-scale generation of adversarial objects through text, offering a broader optimization space beyond geometric tuning. These objects are also interpretable and provide better stealth.

### 2.2 3D CONTENT GENERATION AND PROMPT-GUIDED ATTACKS.

**3D generation paradigms.** Recent advances in 3D generation Poole et al. (2022); Lin et al. (2023); Michel et al. (2022); Liu et al. (2023b); Xu et al. (2024); Shi et al. (2024); Team (2024) have enabled high-quality 3D content synthesis from high-level inputs such as text or images, moving beyond sensor-level or manually crafted inputs. These pipelines typically combine prompt conditioning with multi-view 3D reconstruction using implicit (e.g., NeRF) or explicit (e.g., Gaussian Splatting, triangle mesh) representations, producing editable 3D outputs with high controllability and realism. While primarily developed for benign use, such capabilities open new avenues for adversarial manipulation, where prompts can be optimized to synthesize physically realizable 3D objects that deceive perception systems.

**Adversarial attacks via 3D generation.** While prior works have explored 3D adversarial attacks using generative techniques, most rely on mesh-based representations. Following the seminal work in Athalye et al. (2018), methods (Oslund et al., 2022; Suryanto et al., 2022; 2023; Wang et al., 2022; Zeng et al., 2019) have focused on texture-based perturbations, such as modifying vertex colors

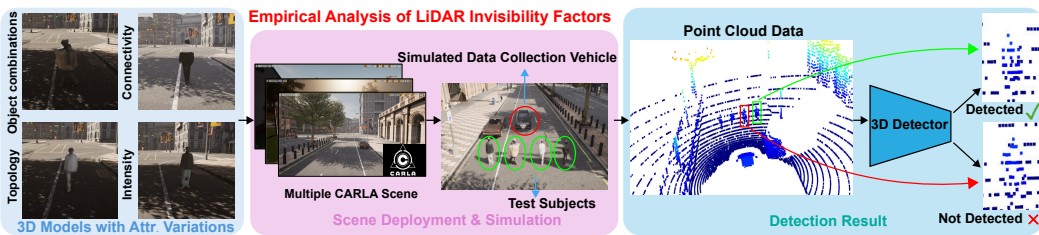

Figure 2: An overview of our empirical analysis pipeline for studying LiDAR invisibility factors. We manipulate the object attributes of 3D pedestrian models or human-object compositions, insert them into diverse CARLA scenes to collect data (middle), and finally evaluate their detectability using LiDAR-based 3D detectors (right).

(Oslund et al., 2022; Xiao et al., 2019; Zeng et al., 2019) or optimizing texture maps on predefined meshes (Suryanto et al., 2022; 2023; Wang et al., 2022). However, these approaches are mainly limited to digital settings and fine-tune colors and textures, lacking evaluation under LiDAR-based perception.

In contrast, we propose adversarially optimizing discrete textual prompts to explore geometric variations, maximizing impact on 3D LiDAR detectors and enabling gradient-driven generation of realizable adversarial objects.

## 3 EMPIRICAL ANALYSIS & MOTIVATION

To thoroughly understand the vulnerabilities of LiDAR-based 3D detectors, we first conduct an empirical study on the detectability of pedestrians in CARLA simulation environment. The test subjects are manipulated 3D pedestrians models, and we collect LiDAR data using a simulated car within CARLA, which allows for high-fidelity simulation across diverse and realistic conditions, including variations in weather, lighting, and scenes.

Prior studies have shown that geometric and physical properties—such as shape (Möller & Trumbore, 2005; Tu et al., 2020; Lee et al., 2020), structural coherence (Xiao et al., 2019), and intensity (Park et al., 2024)—can significantly impact LiDAR-based perception. Therefore, we focus on *modifications to individual objects, including changes in structure, appearance, and combinations of multiple objects*. Specifically, we alter the structure by modifying the pedestrian's topology or adjusting its connectivity to simulate different types of occlusion. Additionally, we modify the pedestrian's appearance in LiDAR by adjusting its intensity. For object combinations, we simulate various human-object compositions, such as boxes and umbrellas. Visualizations of these attribute variations are provided in Appendix Fig. D(d-f). As shown in Fig. 2, our empirical study comprises three stages, providing a realistic assessment of how various factors affect the detection of generated objects.

Beyond object-level attributes, we also explore the *impact of scene-level factors, including the object's viewing angle, distance to the ego vehicle, and ego velocity*, on LiDAR detection outcomes. These variables reflect real-world driving scenarios and are crucial for understanding the robustness of detectors to adversarial attacks on generated objects in dynamic environments. We formalize the relationship between these factors and detection performance using the following decomposition:

$$\text{Detectability:} \gamma = \mathcal{D}\left(\text{Obj}\big(\tilde{\text{Ped}} \cup \text{AddObj}\big), \ \text{Env}(\mathcal{A}, \mathcal{D}, \mathcal{V})\right), \tag{1}$$

where $\gamma$ is the confidence of a 3D object detector $\mathcal{D}$ of the generatd object, with $\tilde{\text{Ped}}$ denoting the constructed 3D Pedestrian based on own topology $\mathcal{T}$, connectivity $\mathcal{C}$, intensity $\mathcal{I}$ and other object combinations AddObj. Topology $\mathcal{T}$, connectivity $\mathcal{C}$, and intensity $\mathcal{I}$ are controlled through a matrix transformation applied to the 3D Pedestrian, either as a masking operation or by adjusting the intensity values. Env$(\mathcal{A}, \mathcal{D}, \mathcal{V})$ encodes the scene context, with $\mathcal{A}$, $\mathcal{D}$, and $\mathcal{V}$ representing the viewing angle, object distance, and ego velocity, respectively. These parameters, including the object's placement angle, the simulation vehicle's position, and the vehicle's speed, are controlled within the CARLA

Table 1: Detection success rates (%) of different LiDAR-based detectors under various object attributes and scene configurations.

| | Single Pedestrian | | Ped. w/ Obj. combinations | Point-based | | Voxel-based | | Point-voxel-based | |
| --- | --- | --- | --- | --- | --- | --- | --- | --- | --- |
| | Structure | Appearance | | | | | | | |
| | Topology / Connectivity | Intensity | | PointRCNN / IA-SSD | | SAFDNet / VoxelNeXt | | HEDNet / PDV | |
| **Scene 1** | | | | 89.6 | 83.6 | 91.0 | 83.1 | 87.0 | 81.3 |
| | Topology ✓ | | | 52.1 | 23.4 | 83.6 | 82.9 | 83.3 | 81.0 |
| | Connectivity ✓ | | | 82.4 | 70.8 | 83.3 | 73.4 | 79.8 | 67.0 |
| | | Intensity ✓ | | 73.6 | 72.9 | 84.3 | 73.1 | 75.7 | 69.5 |
| | | | ✓ | **37.6** | **31.7** | **36.1** | **26.2** | **30.1** | **21.1** |
| **Scene 2** | | | | 92.5 | 84.7 | 95.3 | 87.8 | 86.4 | 76.4 |
| | Topology ✓ | | | 51.3 | 15.9 | 90.2 | 87.6 | 84.1 | 75.8 |
| | Connectivity ✓ | | | 77.3 | 70.1 | 81.7 | 72.8 | 74.1 | 61.5 |
| | | Intensity ✓ | | 74.2 | 64.0 | 89.4 | 72.0 | 73.5 | 61.4 |
| | | | ✓ | **30.8** | **25.0** | **21.8** | **16.4** | **17.7** | **9.9** |
| **Scene 3** | | | | 90.1 | 83.2 | 96.3 | 83.0 | 87.2 | 83.6 |
| | Topology ✓ | | | 47.7 | 26.1 | 88.7 | 83.0 | 84.8 | 83.6 |
| | Connectivity ✓ | | | 83.3 | 65.5 | 90.5 | 63.5 | 72.7 | 66.0 |
| | | Intensity ✓ | | 71.2 | 61.6 | 87.0 | 61.4 | 68.3 | 61.6 |
| | | | ✓ | **36.4** | **36.2** | **32.0** | **21.8** | **18.7** | **11.4** |

environment. This formulation captures how both object attributes and scene conditions jointly influence the detectability of the object.

Our quantitative results in Tab. 1 reveals important insights into how attribute modifications affect adversarial attacks on LiDAR-based 3D detectors. ❶ Object combinations (AddObj) have the greatest impact on Obj($\cdot$) detectability $\gamma$, as combining multiple objects causes a more significant change in the target's point cloud distribution, thereby affecting LiDAR detection results. In Appendix A.1, we further explore the impact of different regions of the pedestrian. ❷ Topology ($\mathcal{T}$), Connectivity ($\mathcal{C}$), and Intensity ($\mathcal{I}$) have a moderate attack effect on LiDAR 3D detectors, as changes to these attributes can alter the object point cloud distribution to some extent. However, these attribute variations do not generalize across detectors, with some detectors being unaffected by attribute variations (e.g., Intensity).

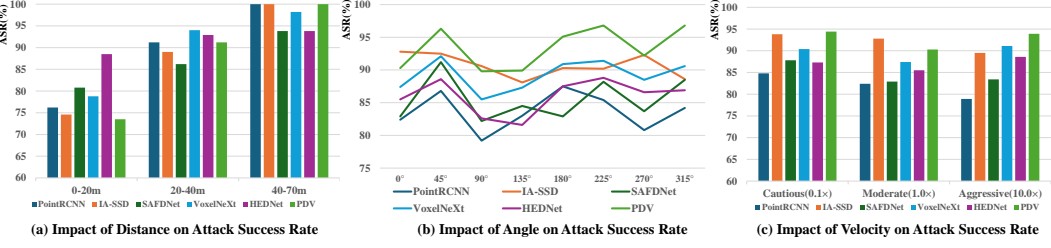

(a) Impact of Distance on Attack Success Rate    (b) Impact of Angle on Attack Success Rate    (c) Impact of Velocity on Attack Success Rate

Figure 3: Effects of environmental factors on 3D adversarial object detectability.

We also gain several insights into the robustness of the attack effect of multiple object combinations under environmental factors, as shown in Fig. 3. ❶ The attack success rate varies significantly with distance ($\mathcal{D}$), indicating that this combinations are more effective at certain distances, challenging the attack's robustness across varying distances. ❷ Unlike simple point addition or removal, adversarially generated objects exhibit viewpoint and motion robustness due to the way Obj($\cdot$) responds to changes in $\mathcal{A}$ and $\mathcal{V}$.

## 4 METHODOLOGY

Motivated by our empirical analysis, we introduce Phy3DAdvGen—a discrete prompt-based framework that generates physically realizable 3D objects capable of evading LiDAR-based detectors in Sec. 4.1 through an end-to-end optimization process over the verb-object-pose (VOP) textual prompt representation. Moreover, to ensure real-world feasibility, we propose to build a physical object pool as the physical props, which allows us to optimize the combination strategy of human subjects and the physical props for the adversarial object generation. As a result, we can conduct physical implementation following the optimized combination strategy.

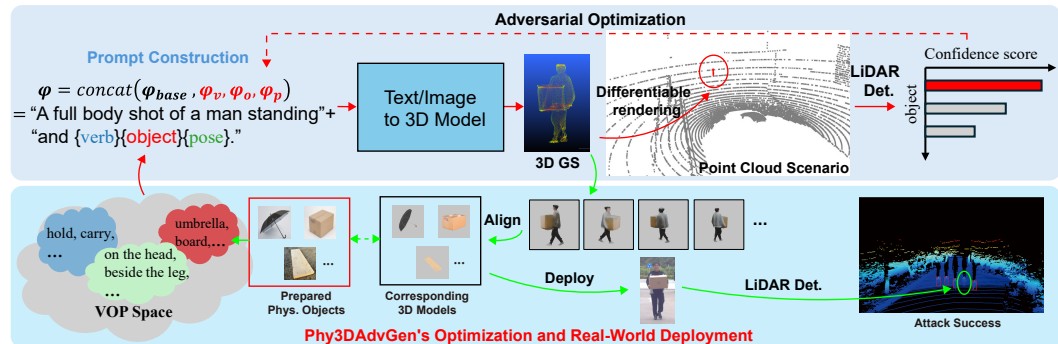

Figure 4: Overview of the Phy3DAdvGen framework: Prompt triplets (verb, object, pose) are optimized end-to-end by backpropagating an adversarial loss from a downstream detector. These prompts guide a text-to-3D model to generate human-object compositions, which are rendered into LiDAR scenes using differentiable rendering. Successful adversarial configurations are physically aligned with real LiDAR sensors for real-world validation.

## 4.1 PHYSICALLY-INFORMED TEXT-TO-3D ADVERSARIAL GENERATION

**Overview.** We first build a verb-object-pose (VOP) textual prompt representation $\mathcal{P}$, which can interpret the target object we want to generate in terms of different verbs ($\mathcal{P}_v$), objects ($\mathcal{P}_o$), and poses ($\mathcal{P}_p$). Our method aims to find a specific textual prompt in $\mathcal{P}$, which can drive a pre-trained text-to-3D generation model $\mathcal{G}$ to generate an adversarial 3D object that the LiDAR-based 3D object detector cannot detect. To this end, we leverage the detection confidence from a downstream detector as the adversarial objective function to guide the optimization in $\mathcal{P}$.

**Verb-object-pose (VOP) textual prompt representation.** We define a discrete verb-object-pose (VOP) space composed of three interpretable subsets: verbs ($\mathcal{P}_v$), objects ($\mathcal{P}_o$), and poses ($\mathcal{P}_p$). The verbs could include "hold", "carry", "push", *etc.*; the objects could include "umbrella","box", "board", *etc.*; and the poses could include "on the head", "in front of the body", "on the back", *etc.* The full prompt space is detailed in Appendix Sec. A.2. We sample a prompt from each subset and form a triplet, which can be represented as ($\varphi_v \in \mathcal{P}_v, \varphi_o \in \mathcal{P}_o, \varphi_p \in \mathcal{P}_p$). Then, we can append the triplet to a base description $\varphi_{\text{base}}$ (*e.g.*, "a full body-shot of a person standing.") to form a complete prompt $\varphi \in \mathcal{P}$ (*e.g.*, "a full body-shot of a person standing and and holding an umbrella on the head."):

$$\varphi = \text{concat}(\varphi_{\text{base}}, \varphi_v, \varphi_o, \varphi_p), \text{subject to}, \varphi_v \in \mathcal{P}_v, \varphi_o \in \mathcal{P}_o, \varphi_p \in \mathcal{P}_p, \quad (2)$$

where concat denotes text-level concatenation of natural language components. We can feed the resulting prompt $\varphi$ to a text-to-3D model $\mathcal{G}$ to generate a 3D human-object instance. The key problem is how to conduct an effective optimization in the space $\mathcal{P}$ that depends on the three subsets to find a $\varphi$, with which the generated 3D object can mislead 3D object detectors. In the following, we introduce the optimization method and the corresponding objective function.

**Detector-aware adversarial objective.** To find an effective prompt in the space $\mathcal{P}$, we conduct end-to-end optimization guided by the LiDAR-based 3D detector. Specifically, we propose the following objective function:

$$\arg\min_{\varphi \in \mathcal{P}} \mathcal{L}_{\text{adv}}(\mathcal{D}(\mathbf{P}), \mathcal{Y}), \text{s.t.}, \mathbf{P} = \mathcal{S}(\mathcal{G}(\varphi), \varepsilon), \quad (3)$$

where $\mathcal{G}(\cdot)$ is a text-to-3D generation model, and $\mathcal{G}(\varphi)$ is to generate a 3D object according to the prompt $\varphi$. $\mathcal{S}(\mathcal{G}(\varphi), \varepsilon)$ is to embed the generated object into the environment $\varepsilon$ and produces a 3D point could $\mathbf{P}$. Note that, the environment $\varepsilon$ is the 3D point cloud of an existing environment. $\mathcal{D}(\mathbf{P})$ is a LiDAR-based 3D object detector to detect the objects in the $\mathbf{P}$ and gets $\{(\mathbf{b}_i, c_i)\}$, where $\mathbf{b}_i$ is the bounding box of detected object and $c_i$ is the detection confidence. We also have $\mathcal{Y} = \{\mathbf{b}^*\}$ as the set of ground-truth bounding boxes. We define the adversarial loss function as

$$\mathcal{L}_{\text{adv}}(\mathcal{D}(\mathbf{P}), \mathcal{Y}) = \max\left(\max_i \left\{c_i \cdot \mathbb{1}_{\text{IoU}(\mathbf{b}_i, \mathbf{b}^*) > \delta}\right\}, \eta\right), \quad (4)$$

where the indicator $\mathbb{1}_{\mathrm{IoU}(\mathbf{b}_i, \mathbf{b}^*) > \delta}$ selects boxes with an IoU above a threshold $\delta$ (*e.g.*, 0.5). If no match exists, the loss falls back to a constant $\eta = 0.1$, ensuring continuous gradient flow. This formulation supports end-to-end optimization of discrete prompts to suppress detection.

**Latent-space optimization.** To realize the above optimization, we first collect words as verbs, objects, and poses and build three subsets (*i.e.*, $\mathcal{P}_v, \mathcal{P}_o, \mathcal{P}_p$). Then, the space $\mathcal{P}$ could be approximated by linearly combining all possible triplets in $(\varphi_v, \varphi_o, \varphi_p) \in \mathcal{P}_v \times \mathcal{P}_o \times \mathcal{P}_p$. To realize feasibility, we conduct the optimization in the latent space. Specifically, we can obtain $N = |\mathcal{P}_v| \cdot |\mathcal{P}_o| \cdot |\mathcal{P}_p|$ prompts via Eq. (2). For each prompt, we can calculate the corresponding latent embedding via a frozen text encoder used in text-to-3D model and obtain a tensor (*i.e.*, $\mathbf{E}_i \in \mathbb{R}^{T \times d}$) for the $i$th prompt, where $T$ is the maximum token length and $d$ is the embedding dimension. Then, the latent embedding of one prompt in $\mathcal{P}$ could be approximated as

$$\mathbf{E} = \sum_i^N \mathbf{w}[i]\mathbf{E}_i, \tag{5}$$

where $\mathbf{w} \in \mathbb{R}^{N \times 1}$. With the above formulation, searching for the $\varphi$ in $\mathcal{P}$ becomes how to confirm the weights $\mathbf{w}$. Instead of learning $\mathbf{w}$ directly, we propose to decompose it into three sub-weights, *i.e.*, $\mathbf{w}_v \in \mathbb{R}^{N \times 1}, \mathbf{w}_o \in \mathbb{R}^{N \times 1}, \mathbf{w}_p \in \mathbb{R}^{N \times 1}$ representing the contributions in the verbs, objects, and poses, which also enlarges the learnable parameters, that is, we have

$$\mathbf{w} = \sigma_{\mathrm{Gumbel}}(\mathbf{w}_v) \odot \sigma_{\mathrm{Gumbel}}(\mathbf{w}_o) \odot \sigma_{\mathrm{Gumbel}}(\mathbf{w}_p), \tag{6}$$

where $\sigma_{\mathrm{Gumbel}}(\cdot)$ is the Gumbel-Softmax function (Jang et al., 2016). We can also feed the latent embedding $\mathbf{E}$ into the text-to-3D generator to generate a 3D object instead of feeding the texts. With the above setups, we can reformulate the Eq. (3) as follows

$$\arg\min_{(\mathbf{w}_v, \mathbf{w}_o, \mathbf{w}_p)} \mathcal{L}_{\mathrm{adv}}(\mathcal{D}(\mathbf{P}), \mathcal{Y}), \mathrm{s.t.}, \mathbf{P} = \mathcal{S}(\mathcal{G}(\mathbf{E}), \varepsilon). \tag{7}$$

Intuitively, with Eq.7, given a detector and a generator, we can optimize the weights $(\mathbf{w}_v, \mathbf{w}_o, \mathbf{w}_p)$, which represent the prompt within $\mathcal{P}$ can drive the generator to produce an object (*i.e.*, $\mathcal{G}(\mathbf{E})$), misleading the LiDAR-based detector.

## 4.2 REAL-WORLD DEPLOYMENT STRATEGY

Although showing effectiveness in the simulation world, the generated 3D object can hardly be realized in the real world: The generated 3D objects do not usually exist in the real world, and we can only produce them via 3D printing, which is difficult to print a human-sized object and requires high costs. To achieve real-world deployment, we propose utilizing existing objects and stacking them in a specific pattern to approach the generated object.

Specifically, we collect a set of real objects (*e.g.*, backpack and umbrella) and calculate their 3D models, which form a set denoted as $\mathcal{O}$. Then, the equation can be reformulated as:

$$\arg\min_{\mathbf{w}_{\mathrm{real}}} \mathcal{L}_{\mathrm{adv}}(\mathbf{E} - \sum_{i=1}^M \mathbf{w}_{\mathrm{real}}[i]\mathbf{E}_{\mathcal{O}_i}). \tag{8}$$

In the physical deployment phase, the optimized weights $\mathbf{w}_{\mathrm{real}}$ are used to adjust the positions, rotations, and scales of the real-world objects, combining them to closely approximate the generated adversarial 3D object. By adjusting the contribution of each physical object (determined by $\mathbf{w}_{\mathrm{real}}[i]$), we can replicate the shape and structure of the target adversarial object. Ultimately, the optimized physical object assembly is deployed in the real environment, and its effectiveness is validated by a LiDAR-based detector. If the physical assembly successfully deceives the detector, the attack is considered successful.

## 5 EXPERIMENT

### 5.1 SETUPS

**Digital & Physical Setup.** Our digital experiments in CARLA using sensor configurations the same as the KITTI (Geiger et al., 2012) dataset, collecting data from three different maps. We generate 100

| LiDAR | LiDAR Setting | | | Trajectory types | | Weather Scenarios | | Velocity | |
|-------|------|------|-------|---------|--------------|--------|-------|------|------|
| Detectors | 32-b | 64-b | **128-b** | Lateral | **Longitudinal** | Cloudy | **Sunny** | Fast | **Slow** |
| PointRCNN | 94.3 | 85.2 | 84.6 | 77.5 | 84.6 | 85.3 | 84.6 | 81.4 | 84.6 |
| IA-SSD | 96.2 | 95.0 | 90.3 | 80.9 | 90.3 | 92.8 | 90.3 | 91.5 | 90.3 |
| SAFDNet | 90.6 | 88.4 | 86.7 | 83.2 | 86.7 | 86.5 | 86.7 | 85.1 | 86.7 |
| VoxelNet | 95.0 | 91.1 | 90.6 | 78.2 | 90.6 | 88.9 | 90.6 | 90.3 | 90.6 |
| HEDNet | 92.5 | 88.3 | 85.3 | 80.8 | 85.3 | 82.8 | 85.3 | 86.4 | 85.3 |
| PDV | 93.0 | 85.7 | 81.6 | 72.0 | 81.6 | 83.5 | 81.6 | 82.2 | 81.6 |

Table 2: Attack Success Rate (%) of Different LiDAR-based Detectors under Various Physical Settings. Bold indicates the baseline configuration. -b refers to the number of beams in the LiDAR.

adversarial 3D objects (50 pedestrians with changing their topology, connectivity, intensity and other 50 with different object combinations) for comprehensive detectability analysis. Physical experiments are conducted in an autonomous driving test field with a stationary ego vehicle equipped with a RS-Ruby 128-beam LiDAR. Around 5,000 real-world point cloud frames are collected and used for detector finetuning with annotations using SUSTech POINTs (Li et al., 2020). Five participants perform in outdoor scenes, with one carrying an adversarial object in the Phy3DAdvGen setting. We evaluate ten adversarial poses as the subject walks towards the vehicle.

**Optimization Setup.** We optimize the logits of discrete prompt components (verb, object, pose) using Adam with a learning rate of 1e-4, while keeping the 3D generation model and LiDAR detector frozen. Optimization runs for 300 steps, minimizing the detector's classification confidence on the generated object. All training is performed on four NVIDIA L40 GPUs.

**Attack Models & Metrics.** We evaluate six high-performing 3D object detection models from the OpenPCDet framework Team (2020), including point-based (PointRCNN Shi et al. (2019), IA-SSD Zhang et al. (2022)), voxel-based (SAFDNet Zhang et al. (2024), VoxelNeXt Chen et al. (2023)), and point-voxel-based methods (HEDNet Zhang et al. (2023), PDV He et al. (2022)). These models are pre-trained on the KITTI dataset and fine-tuned across both digital and physical scenarios. Detection Success Rate (DSR) and Attack Success Rate (ASR) are used to assess detection and attack effectiveness. A detection is successful if the predicted bounding box has an IoU exceed 0.5 with the ground truth. An attack is successful if no box is detected, or if the IoU is below 0.1.

## 5.2 EXPERIMENTAL RESULTS IN DIGITAL AND PHYSICAL SCENARIOS

Table 3: Attack success rate (%) comparison against SOTA attack methods on PointRCNN (CARLA, digital setup).

| Method | ASR |
|--------|-----|
| AdvPC Hamdi et al. (2020) | 64.2 |
| NI-FGSM Lin et al. (2020) | 62.9 |
| Tu et al. (2020)* | 79.3 |
| Zheng et al. (2025)* | 68.4 |
| ScAR Lu & Radha (2023) | 47.5 |
| Phy3DAdvGen (Ours) | 94.6 |

Table 4: Attack success rate (%) under different prompt strategies. All methods are optimized on PointRCNN and evaluated on the average transferability across six LiDAR-based detectors.

| Prompt Strategy | ASR (Avg.) |
|-----------------|------------|
| Random Prompt | 71.0 ± 6.6 |
| LatentPerturb | 78 ± 11.2 |
| Phy3DAdvGen (verb-only) | 62.8 ± 4.7 |
| Phy3DAdvGen (verb + object) | 77.3 ± 8.8 |
| Phy3DAdvGen | 87.7 ± 7.9 |

**Physical Deployment Evaluation.** As shown in Tab. 2, ASR is sensitive to the number of LiDAR beams, with attack success decreasing as the beam count increases from 32-b to 128-b. The ASR drops the most for PDV, from 93.0% with 32-b to 81.6% with 128-b, indicating that fewer beams reduce detection capability, making the attack more effective. ASR also varies with motion type, with longitudinal motion achieving higher success rates than lateral motion, suggesting that lateral movement is more resistant to attacks. Our method demonstrates robustness to weather and velocity changes, as evidenced by consistent ASR values of 85.3% in cloudy conditions and 84.6% in sunny conditions on PointRCNN, as well as 90.3% at fast velocities and 90.6% at slow velocities on VoxelNet. Some of our findings in real-world validation share similarities with those in simulation, such as the impact of environmental factors like vehicle speed and weather.

**Comparison with SOTA Attack Methods.** Tab. 3 compares Phy3DAdvGen with representative adversarial baselines under the digital CARLA setup using the PointRCNN detector. The first two methods—AdvPC and NI-FGSM—perturb LiDAR point clouds directly, achieving moderate attack success rates (ASRs) of 62.9–64.2%, as the perturbations require manipulation of the vehicle's sensor point cloud data, making physical deployment challenging. To provide a more geometry-aware comparison, We reimplement Tu et al. (2020)* and Zheng et al. (2025)*, finding that their performance is more effective than fine-tuning the original point clouds. However, since they optimize the geometry of the 3D object, their optimization space remains limited. In contrast, Phy3DAdvGen optimizes discrete prompts to generate realistic and undetectable 3D human-object compositions, achieving a significantly higher ASR of 94.6%. Semantically rich prompts offer better interpretability, making real-world deployment easier, while providing greater optimization potential and enabling the generation of diverse object compositions that can impact LiDAR detection.

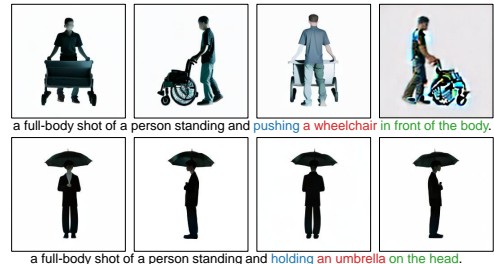

a full-body shot of a person standing and pushing a wheelchair in front of the body.

a full-body shot of a person standing and holding an umbrella on the head.

Figure 5: Multi-view images of the 3D adversarial object generated by Phy3DAdvGen, with prompts below.

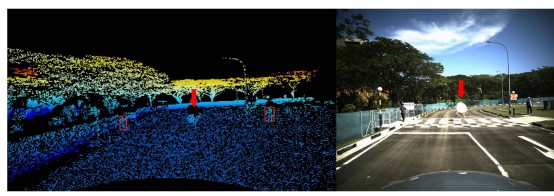

Figure 6: Visualization demonstrating the success of the Phy3DAdvGen attack on the SAFDNet LiDAR detector in a real-world deployment scenario.

**Effect of Prompt Strategy.** Tab. 4 summarizes the ASR performance under different prompt strategies. LatentPerturb achieves the highest ASR across detectors (78.0%), benefiting from continuous-space optimization. However, it lacks semantic interpretability and physical deployability, as the perturbations are non-interpretable and impractical for real-world deployment. In contrast, Phy3DAdvGen optimizes prompts in a discrete, interpretable space (verb, object, pose), yielding realistic 3D human-object compositions. It achieves the highest average ASR, demonstrating strong cross-detector generalization. Ablation results reveal that optimizing only the verb provides limited effects over random prompts, while jointly optimizing the verb and object significantly boosts ASR (77.3%) and further elevates it to 87.7%, underscoring the importance of object semantics and their combinations in enhancing adversarial effectiveness.

**Visualization.** Fig. 5 demonstrates multi-view images of an adversarial object generated by Phy3DAdvGen, showcasing consistency across multiple views and a physically realizable multi-object composition based on the given prompts. We also present some failure cases in Appendix A.3. Additionally, Fig. 6 demonstrates the effectiveness of our physical deployment method, where the adversarial pedestrian walking on the zebra crossing was undetected by the detector, posing a significant danger. This failure to detect the adversarial object poses a significant danger, underscoring the vulnerability of current LiDAR-based 3D detectors to adversarial attacks.

## 6 CONCLUSION

In this work, we present the first empirical study in the CARLA simulation environment to investigate factors affecting the detectability of 3D objects in LiDAR detectors. We find that attribute variations of a single object do not form a generalizable attack across LiDAR detectors, whereas combinations of multiple objects have a more powerful effect on most LiDAR detectors. Based on these insights, we propose Phy3DAdvGen, a physically realizable text-to-3D adversarial generation framework that controls these combinations to generate LiDAR-evasive 3D human-object compositions. Extensive experiments in both simulation and real-world scenarios demonstrate the method's effectiveness across multiple SOTA detectors, highlighting the need for more robust LiDAR detectors against generative threats.

# 7 ETHICS STATEMENT

We affirm that this study adheres to the ICLR Code of Ethics, and all work has been conducted in compliance with ethical guidelines. The research does not involve human subjects or animals, and no privacy concerns or legal issues were encountered. Additionally, the research does not raise any conflicts of interest or discrimination concerns. The use of LiDAR data and 3D generation techniques was done following relevant safety and ethical standards.

# 8 REPRODUCIBILITY STATEMENT

To ensure the reproducibility of our results, we provide detailed descriptions of our methodology, experimental setups, and data processing steps in the main text and supplementary materials. Our source code, 3D models, and data processing scripts will be made publicly available through an anonymous repository to facilitate independent validation of our results. For clarity, we also provide clear explanations of any assumptions made in the experiments and relevant parameters for reproducing the adversarial generation results.

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

# A APPENDIX

In this section, we provide additional analyses, including the impact of body-part occlusions on LiDAR-based pedestrian detection, a discussion of our discrete prompt space for 3D generation, and visual comparisons of LGM, Meshy, and Shap-E. We also highlight limitations and future directions, focusing on spatial grounding, stochastic generation, and multi-sensor integration. Additionally, the statement regarding the use of Large Language Models (LLMs) is included at the end.

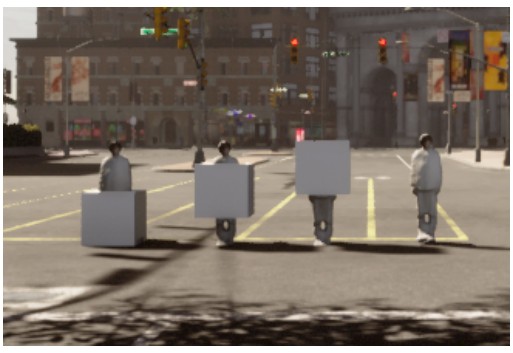

Figure A: Visualization of simulated body-part occlusions. From left to right: feet occlusion, torso occlusion, head occlusion, and unoccluded subject.

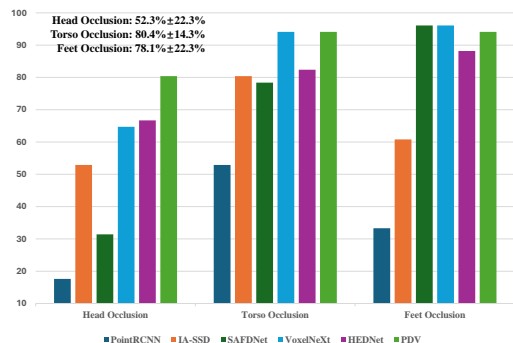

Figure B: Attack success rates (%) for different body-part occlusions across six LiDAR-based 3D detectors. Bold numbers (top-left) indicate mean and standard deviation per occlusion type.

## A.1 OCCLUSION ANALYSIS BY BODY REGION

As an extension of our empirical study, we report an additional occlusion-based experiment that investigates how occlusions of specific body regions affect LiDAR-based pedestrian detection. As shown in Fig. A, we simulate selective occlusions by placing geometric cubes over the feet, torso, and head regions of human subjects in the CARLA environment, allowing us to isolate each region's contribution to 3D detector performance.

The quantitative results are shown in Fig. B, where the attack success rate is reported for each occlusion type across six state-of-the-art LiDAR-based detectors. The bolded statistics in the top-left corner summarize the average attack success rate and standard deviation across detectors for each occlusion region. Notably, torso occlusion leads to the highest average attack success rate (80.4%) with relatively low variance, indicating that the torso contains critical structural features essential for reliable detection. Feet occlusion also causes significant degradation (78.1% average success rate), likely due to the loss of ground-contact cues in point clouds. In contrast, head occlusion results in much lower attack success (52.3%) and higher variability, suggesting that the head contributes less to LiDAR-based detection, and detectors are more robust to its absence.

These findings highlight the unequal contributions of body regions to 3D perception, with the torso and feet being more crucial than the head. This motivates our use of spatially guided prompts to control object placement—favoring lower-body occlusions to improve adversarial stealth with minimal visual footprint.

## A.2 DISCRETE PROMPT SPACE AND 3D GENERATION

This section complements the main paper by providing additional visualizations and analysis of our discrete prompt space for controllable 3D generation. As shown in Fig. C, the left panel presents the verb-object-pose (VOP) prompt pool, where each prompt is formed by selecting one token from each of the three categories. This compositional design enables fine-grained control over human-object interactions and facilitates diverse, semantically meaningful generations. The right panel of Fig. C shows representative multi-view images generated from sampled triplets. Examples such as "holding an umbrella above the head" or "pushing a wheelchair in front of the body" demonstrate how semantic modularity leads to rich and plausible physical configurations. The discrete nature of this space

| Verb Pool | Object Pool | Pose Pool |
|---|---|---|
| hold, carry, place, tote, hug, wear, push | umbrella, box, board, stick, guitar case, bag pack, shovel, folding chair, balloon, soccer ball wheelchair, teddy bear, swimming ring | above the head, on the head, in front of the body, over the shoulder, on the back, on the lap, beside the leg, on the ground |

a full-body shot of a person standing and pushing a wheelchair in front of the body.

a full-body shot of a person standing and holding an umbrella on the head.

Figure C: Visualization of the discrete prompt space and corresponding 3D outputs. Left: Verb-object-pose (VOP) prompt pool. Right: Multi-view 3D generations illustrating compositional controllability from sampled triplets.

further supports efficient exploration and adversarial prompt optimization, allowing us to discover configurations that are both physically realizable and evasive to LiDAR-based detectors.

Overall, these visualizations reinforce the main claim that compositional prompts enable interpretable and controllable adversarial generation, providing a flexible and scalable mechanism for producing diverse 3D attack candidates.

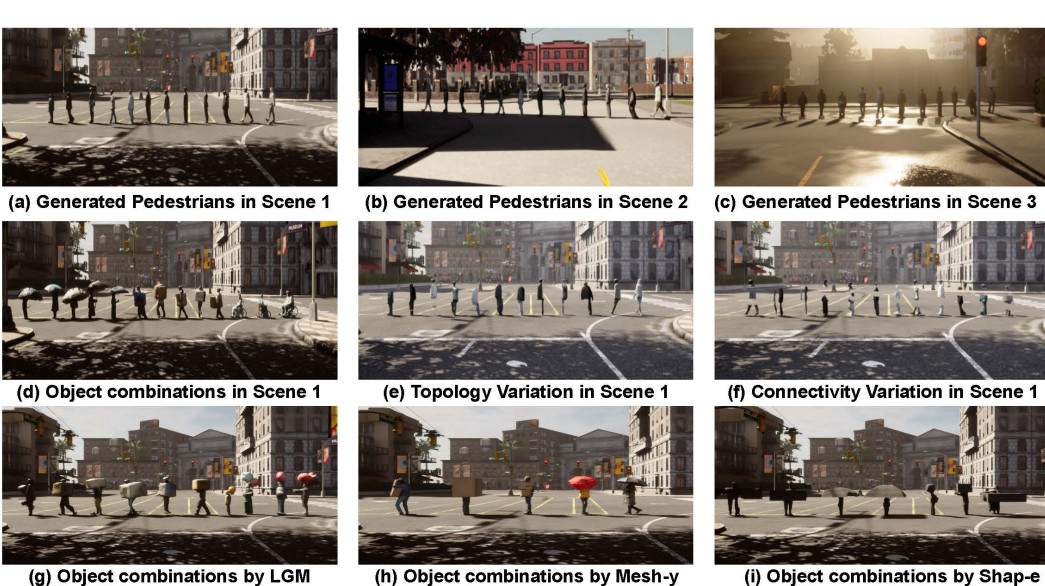

(a) Generated Pedestrians in Scene 1    (b) Generated Pedestrians in Scene 2    (c) Generated Pedestrians in Scene 3

(d) Object combinations in Scene 1    (e) Topology Variation in Scene 1    (f) Connectivity Variation in Scene 1

(g) Object combinations by LGM    (h) Object combinations by Mesh-y    (i) Object combinations by Shap-e

Figure D: Visualization of 3D pedestrians and attribute ablations in CARLA. Subfigures (a–c) depict 3D pedestrians across various scenes (Scene 1 - Scene 3), while (d–f) showcase object-level attribute ablations. (g) LGM results under diverse prompts, (h) Meshy-generated instances, and (i) Shap-E outputs.

## A.3 PROMPT-CONDITIONED GENERATION: VISUAL COMPARISONS AND FAILURES

Beyond prompt design, we provide qualitative comparisons to explore how different text-to-3D backbones interpret prompt semantics, with a focus on generation consistency and common failure patterns.

We begin by visualizing the outputs of LGM, Meshy, and Shap-E under identical textual prompts, rendered within simulated urban scenes using the CARLA environment (Fig. D). Subfigure (g) shows representative LGM-generated samples conditioned on diverse prompts, demonstrating strong semantic alignment and consistent object placement. Subfigure (e) presents outputs from Meshy, a commercial closed-source model known for its high photorealism. Subfigure (f) displays results

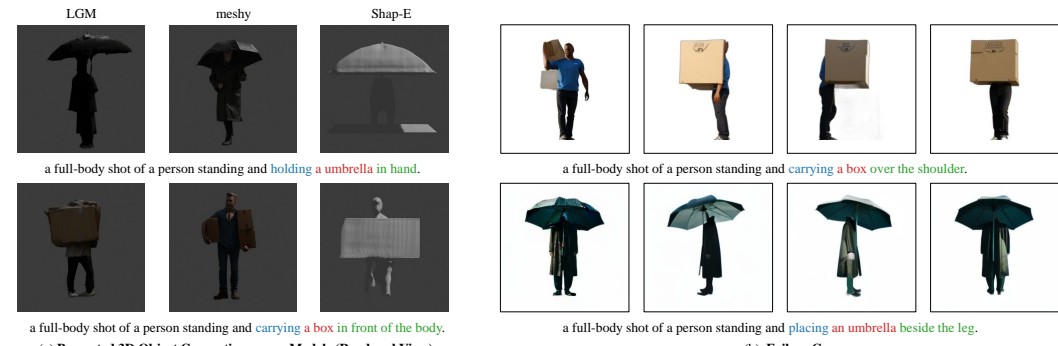

| LGM | meshy | Shap-E |

a full-body shot of a person standing and holding a umbrella in hand.

a full-body shot of a person standing and carrying a box over the shoulder.

a full-body shot of a person standing and carrying a box in front of the body.

a full-body shot of a person standing and placing an umbrella beside the leg.

(a) **Prompted 3D Object Generation across Models (Rendered View)**

(b) **Failure Cases**

Figure E: Prompt-conditioned 3D generation and failure analysis. (a) Rendered views from LGM, Meshy, and Shap-E under identical prompts. (b) Failure cases from LGM: the top row shows inconsistent multi-view geometry; the bottom row shows spatial misalignment, where generated objects appear but are misplaced relative to the prompt instructions.

from Shap-E, which is characterized by greater geometric diversity but weaker adherence to prompt conditioning. These qualitative comparisons reveal characteristic trade-offs among the three models in terms of realism, diversity, and semantic controllability.

To further examine failure modes in prompt-conditioned generation, Fig. E(a) presents comparisons across LGM, Meshy, and Shap-E under identical prompts. The outputs differ notably in geometry, object placement, and overall fidelity. Meshy achieves high photorealism, but as a commercial closed-source model, it lacks controllability and transparency. Shap-E, while producing diverse structures, often yields implausible or structurally invalid results. In contrast, LGM offers a favorable trade-off between prompt adherence and visual realism, making it our primary choice for controlled generation.

Fig. E(b) highlights representative failure cases from LGM. The top row shows inconsistencies across viewpoints, resulting in unrealistic multi-view geometry. The bottom row illustrates spatial grounding errors—for example, generating an umbrella far from the leg despite the prompt's instruction. These observations motivate future improvements in both spatial understanding and semantic controllability, even for the relatively more stable LGM.

## A.4 MODEL LIMITATIONS AND FUTURE DIRECTIONS

Building on the above observations, we summarize key limitations of current text-to-3D pipelines. Despite the flexibility offered by compositional prompts, challenges persist in both semantic controllability and spatial grounding. First, while our framework searches over spatially meaningful prompts, existing backbones often struggle with executing complex spatial relations (e.g., placing an open umbrella in front of the body to obscure the legs), limiting precise grounding. Second, although verb-object-pose compositions support diverse generations, their effectiveness heavily depends on the semantic coverage of the pre-trained backbone—rare or unseen combinations can lead to implausible results. Third, the generation process is inherently stochastic; diffusion- or mesh-based pipelines may yield inconsistent outputs even under identical prompts, impacting reliability. Finally, in the absence of explicit spatial supervision, achieving fine-grained physical realism remains a significant challenge.

Moreover, our current approach focuses solely on LiDAR-based detectors via geometry-level manipulations. However, modern autonomous systems typically integrate both LiDAR and RGB cameras for robust 3D perception. Since our generated 3D assets also contain texture components, future work could explore jointly optimizing shape and appearance to fool both LiDAR and vision-based detectors. This direction may lead to more realistic, transferable, and comprehensive multi-sensor adversarial scenarios.

## A.5 THE USE OF LARGE LANGUAGE MODELS

In this paper, we utilized a Large Language Model primarily for proofreading and refining the manuscript. The model assisted in improving language clarity, grammar, and overall writing quality. All content and ideas presented in this paper are the authors' own, and the LLM's role was limited to language enhancement.

