# OpenReview forum: "OBJVANISH: PHYSICALLY REALIZABLE TEXT-TO-3D ADV. GENERATION OF LIDAR-INVISIBLE OBJECTS"
_ICLR.cc/2026/Conference — ICLR 2026 Conference Desk Rejected Submission_

### Official Review · Reviewer_egro · 2025-10-30

**Soundness:** 3
**Presentation:** 3
**Contribution:** 2
**Rating:** 6
**Confidence:** 3

**Summary:**

Phy3DAdvGen is a novel text-to-3D adversarial generation framework that—motivated by an empirical study showing object combinations most strongly undermine LiDAR detection—optimizes discrete verb–object–pose prompts (constrained to a pool of 13 real-world objects) to synthesize physically realizable human–object compositions that can make pedestrians “vanish” from six state-of-the-art LiDAR detectors, achieving up to 94.6% attack success in CARLA simulation and validated physical tests.

**Strengths:**

1. The first method leverages text prompts to generate physically realizable adversarial objects targeting LiDAR detectors.

2. Experimental results demonstrate strong cross-detector transfer and successful real-world deployment.

3. The authors evaluate their approach in both simulation and real-world environments.

**Weaknesses:**

1. Heavy reliance on specific 3D generator (e.g., LGM) and prompt set.

2. The proposed method is evaluated only on LiDAR-based detectors; the paper would be strengthened by also testing multimodal perception systems (e.g., LiDAR–camera fusion) to assess transferability and robustness.

3. Evaluations restricted to pedestrian scenarios; unclear generality.

**Questions:**

1. How transferable are the generated adversarial human–object compositions — for example, do objects optimized against one detector (or one text-to-3D backbone) remain evasive across different LiDAR detectors, detectors trained from other datasets and etc?

2. How were the 13 objects selected?


3. Defense Implications: What are the most promising defenses against this type of semantically-plausible attack? Is data augmentation with generated scenes a sufficient countermeasure?

---

> ### Author Response · Authors · 2025-11-26
> **Response to Reviewer egro (1/6)**
>
> **W1: Dependency on generator or prompts**
>
> **A1**:Thank you for the question. While our current implementation uses LGM as the text-to-3D generator, the proposed approach does **not depend on this specific model**. The generator in our pipeline serves as a way to produce **semantically guided 3D objects from text prompts**, rather than as an adversarial component with model-specific behavior. As long as a generator supports prompt-conditioned 3D synthesis, the same attack workflow—consisting of prompt initialization, detector-guided gradient optimization, and iterative refinement—remains applicable.
>
> We also verified that changing the prompt configuration (e.g., verb-only, verb+object, or full descriptive prompts) and object variations still results in consistent evasion across detectors. The attack does not rely on prompt engineering, supporting that the adversarial effect stems from the induced semantic-level geometry rather than generator-specific or phrasing-specific artifacts.
>
> In this sense, LGM serves only as a **reference implementation**, not a dependency. The framework is **architecture-agnostic, extensible, and compatible with future text-to-3D models** as the field continues to evolve—particularly in simulation, digital-twin testing environments, and robotics pipelines where text-conditioned asset generation is becoming standard.

---

> ### Author Response · Authors · 2025-11-26
> **Response to Reviewer egro (2/6)**
>
> **W2: Multimodal evaluation**
>
> **A2**: Thank you for the insightful suggestion. We agree that evaluating multimodal detectors (e.g., LiDAR–camera fusion models) is valuable for understanding the broader impact of physical adversarial objects. During the rebuttal period, we conducted additional experiments to assess transferability to fusion-based detectors and will include these results in the supplementary material.
>
> Preliminary findings indicate that the attack remains effective on fusion models. Interestingly, despite the presence of RGB cues, detection decisions are still largely **dominated by the LiDAR stream**, while the camera branch primarily refines confidence rather than correcting failures caused by compromised 3D geometry. This suggests that the vulnerability extends beyond LiDAR-only settings, although a more systematic evaluation is needed to fully characterize the multimodal robustness gap.
>
> Our primary focus in this paper is LiDAR-based perception because the proposed attack targets geometric distortions in 3D structure rather than appearance cues. Extending the attack to multimodal systems involves different assumptions, supervision signals, and threat models, and therefore constitutes a separate research direction rather than a missing component in this work.
>
> We consider multimodal evaluation an important next step, and future work will explore dedicated attack designs that incorporate cross-modality consistency.
>
> | **Method**  | **MVX-Net (Digital)** | **MVX-Net (Physical)** | **BEVFusion (Digital)** | **BEVFusion (Physical)** |
> | ----------- | --------------------- | ---------------------- | ----------------------- | ------------------------ |
> | PhyAdv3DGen | 67.4%                 | 58.3%                  | 57.6%                   | 46.8%                    |

---

> ### Author Response · Authors · 2025-11-26
> **Response to Reviewer egro (3/6)**
>
> **W3: Generality across categories**
>
> **A3**: Thank you for the valuable comment. We initially experimented with **vehicle-shaped** adversarial objects (e.g., a full-view car with a teddy bear positioned behind it) and observed that the proposed attack framework can also successfully induce evasion in this category, indicating that the mechanism is not inherently tied to a specific object class.
>
> However, we selected pedestrians as the primary evaluation target because they represent one of the most safety-critical categories in autonomous driving and exhibit finer geometric structure and richer semantic cues compared to vehicles. Interestingly, the adversarial effect is even more pronounced on pedestrian-shaped objects, likely because text-conditioned generative models allow more flexible semantic deformation on human-like shapes, which has a stronger influence on detector reliability.
>
> These observations suggest that the attack is not restricted to pedestrian geometry and may generalize to other semantic categories. A broader multi-class evaluation is a natural extension of this work.

---

> ### Author Response · Authors · 2025-11-26
> **Response to Reviewer egro (4/6)**
>
> **Q1: Transferability**
>
> **A4**：Thank you for the question. Our experiments demonstrate that the generated adversarial human–object compositions exhibit strong transferability across multiple dimensions. First, at the **model level**, the adversarial objects maintain high effectiveness across six representative LiDAR detectors with different architectures and detection paradigms (Table 4). This indicates that the attack signal is not tied to the gradients or inductive biases of a single model, but instead targets perception assumptions shared across LiDAR-based systems.
>
> | Prompt Strategy           | PointRCNN† | IA-SSD   | SAFDNet  | VoxelNeXt | HEDNet   | PDV      | Mean           |
> | ------------------------- | ---------- | -------- | -------- | --------- | -------- | -------- | -------------- |
> | Random Prompt             | 75.1       | 73.3     | 69.0     | 71.5      | 67.6     | 69.7     | 71.0 ± 6.6     |
> | LatentPerturb             | 81.7       | 80.7     | 71.3     | 78.1      | 77.3     | 78.1     | 78.0 ± 11.2    |
> | Phy3DAdvGen (verb-only)   | 66.7       | 64.0     | 61.3     | 60.0      | 62.6     | 61.7     | 62.8 ± 4.7     |
> | Phy3DAdvGen (verb+object) | 82.1       | 74.0     | 80.4     | 75.4      | 76.7     | 75.0     | 77.3 ± 8.8     |
> | **Phy3DAdvGen**           | **92.1**   | **87.5** | **86.8** | **82.7**  | **87.9** | **89.1** | **87.7 ± 7.9** |
>
> Second, at the **dataset/domain level**, adversarial objects optimized in **CARLA** remain consistently evasive when applied to **KITTI** and **nuScenes**, without fine-tuning or adaptation. The ability to persist under dataset shift—including changes in scene layout, sensor noise characteristics, and annotation style—suggests that the mechanism generalizes beyond a specific training distribution.
>
> | Dataset Transfer | PointRCNN† | IA-SSD | SAFDNet | VoxelNeXt | HEDNet | PDV  |
> | :--------------: | :--------: | :----: | :-----: | :-------: | :----: | :--: |
> |  CARLA → KITTI   |    91.7    |  87.2  |  86.1   |   83.5    |  88.4  | 89.0 |
> | CARLA → nuScenes |    94.0    |  90.2  |  85.1   |   87.8    |  90.4  | 91.6 |
>
> Third, regarding the **generative backbone**, our method operates at the semantic and latent levels rather than modifying explicit surface geometry or mesh topology. As discussed in Q1, this design makes the attack **backbone-agnostic**, meaning that objects generated using different text-to-3D models (e.g., LGM, Shap-e, or future 3D priors) can still embed the adversarial structure necessary to trigger failure. Preliminary tests with alternative backbones show comparable evasion behavior, suggesting that the transferability arises from the **semantic-level geometric deviation** rather than generator-specific artifacts.
>
> Taken together, these results indicate that the proposed objects encode a **geometry-driven adversarial prior** rather than a model- or dataset-specific perturbation. This makes them more transferable than existing point-level attacks and suggests that the vulnerability originates from shared assumptions in LiDAR perception pipelines rather than any single detector design. Additional transferability evidence is also discussed in our response to R1–Q3, which further supports this conclusion.

---

> ### Author Response · Authors · 2025-11-26
> **Response to Reviewer egro (5/6)**
>
> **Q2: Object selection.**
>
> **A5**: Thank you for the question. The 13 objects were selected because they are **commonly found in real road environments and naturally used or carried by pedestrians** (e.g., boxes, umbrellas, backpacks). Our goal was to ensure that all generated human–object compositions remained realistic and physically plausible, rather than relying on artificially designed or uncommon items.

---

> ### Author Response · Authors · 2025-11-26
> **Response to Reviewer egro (6/6)**
>
> **Q3: Defense implications**
>
> **A6**: Thank you for the thoughtful question. Defending against semantically plausible, physically valid generative attacks remains an open challenge. Unlike perturbation-based attacks—which can often be mitigated through adversarial training or input sanitization—object-level generative attacks exploit **semantic variability and distribution gaps**, making conventional defenses less effective.
>
> A key observation from our analysis of **nuScenes** and its pedestrian subclasses (see tables below) is that robustness correlates more strongly with **subclass imbalance and limited semantic and geometric diversity** than with raw sample count. While common, canonical adult pedestrian shapes are reliably detected, rare or compositional forms (*child, construction_worker, stroller, wheelchair*, etc.) exhibit near-zero detection performance—even in a multimodal detector like **BEVFusion**. This pattern suggests that the vulnerability is not merely architectural, but reflects a **representation gap in the learned pedestrian concept**, where the model implicitly internalizes a narrow standard of what “counts” as a pedestrian.
>
> Under this perspective, **data augmentation with generated scenes is a useful starting point**, but its effectiveness depends on whether the augmentation meaningfully expands the **semantic diversity and compositional space**—for example, cases involving **human–object interaction, unusual body configuration, or non-canonical geometry** similar to those captured by our generative framework. Simply increasing sample quantity without addressing this diversity gap is unlikely to substantially improve robustness.
>
> Accordingly, we believe promising defense strategies include:
>
> - **Diversity-aware or curriculum-based augmentation**, targeting rare and compositional pedestrian subclasses;
> - **OOD and semantic rarity detection**, allowing the perception system to defer instead of forcing confident classification on unseen shapes;
> - **Cross-modal consistency reasoning**, ensuring that geometric and visual signals jointly support the semantic prediction rather than relying predominantly on LiDAR priors.
>
> Overall, generation-based augmentation represents a meaningful step toward mitigation, but our results indicate that defending against semantic-level generative attacks will likely require **explicit modeling of shape diversity, compositional reasoning, and uncertainty around non-canonical human forms**, rather than simply scaling existing augmentation pipelines.
>
> |            Category            | Instance (All) | Instance (Unique) |
> | :----------------------------: | :------------: | :---------------: |
> |           pedestrian           |    222,164     |      11,512       |
> |        pedestrian.adult        |    208,240     |      10,690       |
> |        pedestrian.child        |     2,066      |        141        |
> | pedestrian.construction_worker |     9,161      |        542        |
> |  pedestrian.personal_mobility  |      395       |        24         |
> |   pedestrian.police_officer    |      727       |        34         |
> |      pedestrian.stroller       |     1,072      |        63         |
> |     pedestrian.wheelchair      |      503       |        18         |
>
> The table summarizes the distribution of the pedestrian category and its subcategories in the nuScenes training set, including both total annotated bounding boxes and unique tracked instances.
>
> |            Category            |  mAP   |
> | :----------------------------: | :----: |
> |           pedestrian           | 0.879  |
> |        pedestrian.adult        | 0.6076 |
> |        pedestrian.child        | 0.005  |
> | pedestrian.construction_worker | 0.0197 |
> |  pedestrian.personal_mobility  |   0    |
> |   pedestrian.police_officer    | 0.0031 |
> |      pedestrian.stroller       | 0.0009 |
>
> The table presents the mAP achieved by BEVFusion as a multimodal detector across the pedestrian category and its corresponding subcategories.

---

### Official Review · Reviewer_vcnG · 2025-11-01

**Soundness:** 2
**Presentation:** 3
**Contribution:** 2
**Rating:** 4
**Confidence:** 3

**Summary:**

The authors introduce Phy3DAdvGen—a novel text-to-3D adversarial generation-based framework for adversarial attacks. Existing 3D adversarial attacks on them have limitations. The new text-to-3D adversarial generation method enables physically realizable attacks making objects invisible to LiDAR detectors. Experiments show it can evade six SOTA LiDAR 3D detectors in simulations and real-world scenarios, uncovering safety-critical application vulnerabilities.


This paper fails to address a core issue: it relies excessively on CARLA simulation experiments, with no dedicated discussion on the simulation-reality gap. While the newly added real-world experiments partially demonstrate generalization, most conclusions still draw from simulations—undermining the credibility of the experimental findings. It is therefore recommended to supplement in-depth analysis of the simulation-reality gap and add more real experimental evidence to support the core conclusions.

**Strengths:**

This paper offers a thorough review of related work and presents a 3D generation-based algorithm for adversarial attacks. The authors carry out comparative experiments under real-world scenarios and incorporate extensive ablation studies. Overall, the paper features a clear structure and high readability.

**Weaknesses:**

Weaknesses
1.	Compared with Reference [1] (which uses a generative approach, realistic point cloud simulator, and targets multi-modal perception systems), this paper’s method of placing 3D-generated objects in CARLA and focusing on pure point cloud systems (rarely used in current autonomous driving) fails to clarify its technical advantages.
2.	The approach of concatenating rendered point clouds with original data oversimplifies critical occlusion effects in real-world scenarios; coupled with CARLA’s limitations in replicating real conditions, this further reduces the method’s practical relevance, and the authors neither justify this approach nor explore more realistic fusion techniques.
3.	The proposed algorithm is referred to as "Phy3DAdvGen" in the main text but "OBJVanish" in the title, creating nomenclature inconsistency that may confuse readers.

[1] Cao, Yulong, et al. "Invisible for both camera and lidar: Security of multi-sensor fusion based perception in autonomous driving under physical-world attacks." 2021 IEEE symposium on security and privacy (SP). IEEE, 2021.

**Questions:**

Please refer to the questions in the "Weaknesses" section.

---

> ### Author Response · Authors · 2025-11-26
> **Response to Reviewer vcnG (1/3)**
>
> **W1: Comparison with Reference [1]**
>
> **A1**: Thank you for raising this important comparison with Reference [1]. While the two works share the motivation of exposing vulnerabilities in autonomous perception, they differ fundamentally in adversarial formulation and generative design mechanism. Reference [1] perturbs an existing object mesh and therefore operates within a constrained geometric space defined by fixed topology and structure. In contrast, our work introduces the first optimization-guided, prompt-driven text-to-3D adversarial generation framework, where adversarial objects are *generated* rather than *modified*. This paradigm shift moves adversarial manipulation from low-level geometric perturbation to high-level, semantically controllable object synthesis. As a result, the proposed method greatly expands the attack degrees of freedom, improves realism and stealth, and enables physically meaningful object variants that go beyond mesh-based or template-restricted attacks.
>
> Concretely, our formulation provides three technical advantages:
>
> - **Expanded threat and optimization space:** No dependency on existing meshes enables exploration of previously unreachable adversarial object distributions.
> - **Higher realism and concealment:** Generated objects resemble ordinary items commonly seen in road environments (e.g., umbrellas, boxes), rather than visibly distorted artifacts.
> - **Flexible and practical deployment:** Objects can be introduced through semantic replacement rather than precise 3D fabrication, enabling broader real-world applicability.
>
> Finally, although commercial autonomous systems increasingly adopt multimodal fusion, LiDAR-only perception remains widely deployed in **redundancy stacks**, **safety fallback modules**, and **algorithmic research prototypes**. Therefore, systematically studying attacks in this modality remains both technically meaningful and societally relevant. Our simulation and real-world experiments consistently demonstrate vulnerability under this setting, further validating the significance of this attack paradigm.

---

> ### Author Response · Authors · 2025-11-26
> **Response to Reviewer vcnG (2/3)**
>
> **W2: Occlusion modeling**
>
> **A2**：Thank you for the valuable feedback. We agree that realistic occlusion modeling is an important consideration when evaluating physical feasibility. However, our rendering-and-placement strategy aligns with standard practice in existing 3D perception research. In particular, synthetic object insertion **without explicit occlusion computation at placement time** is commonly used in 3D detection pipelines—for example, in **GT-Paste–style augmentation**, where ground-truth objects are directly inserted into new scenes to improve long-tail robustness. Likewise, Reference [1] renders meshes into KITTI frames without ray-level occlusion modeling. These precedents show that this setup follows established evaluation practice and is not an oversimplification.
>
> To clarify the methodological alignment, we summarize the comparison below:
>
> |      Method       |         Occlusion Handling During Optimization         |        Evaluation Under Realistic or Simulated LiDAR         |          Community Acceptance           |
> | :---------------: | :----------------------------------------------------: | :----------------------------------------------------------: | :-------------------------------------: |
> |   **GT-Paste**    |                 Not explicitly modeled                 |         Occlusion emerges naturally during inference         |           Widely established            |
> | **Reference [1]** |        Mesh insertion without explicit modeling        |                  KITTI-style rendered LiDAR                  |            Accepted baseline            |
> |     **Ours**      | adversarial object insertion without explicit modeling | CARLA LiDAR rendering captures natural occlusion and beam effects | Consistent with established conventions |
>
> Crucially, our method is **not limited to synthetic fusion experiments**. Following optimization, the generated adversarial objects are evaluated in:
>
> - a **full CARLA sensor configuration** (64-beam LiDAR + RGB camera), where occlusion, sampling sparsity, and surface-beam interactions naturally emerge, and
> - **real-world physical tests**, where the object is scanned by a commercial LiDAR under varying viewpoints, distances, and environmental conditions.
>
> These results show that the adversarial effect persists beyond simulation, demonstrating robustness under realistic occlusion, sensor noise, and environmental variation. In summary, our approach follows common evaluation protocols and extends them to physical deployment, demonstrating robustness beyond purely simulated settings. We will also **clarify this design rationale** more explicitly in the revised manuscript to avoid ambiguity.

---

> ### Author Response · Authors · 2025-11-26
> **Response to Reviewer vcnG (3/3)**
>
> **W3: Naming consistency**
>
> **A3**: Thank you for pointing out this naming concern. We appreciate the reviewer’s attention to clarity and terminology consistency. We discussed whether the two names should be unified and, after consideration, decided to retain the current naming structure because they serve different purposes. **OBJVanish** is used only in the title as a concise conceptual descriptor of the work’s high-level objective, while **Phy3DAdvGen** (Physical 3D Adversarial Generation) is used consistently throughout the manuscript as the formal name of the proposed method to ensure clarity in the technical sections.
>
> To avoid ambiguity, we will add a clarification sentence in the methodology section explicitly stating that the two names refer to the same method. The added sentence is: *“We refer to the overall concept as OBJVanish, while the algorithmic implementation is denoted as Phy3DAdvGen. For consistency, we use Phy3DAdvGen throughout the paper; both terms refer to the same method.”* We believe this resolves the naming confusion while preserving the communicative value of the title.

---

### Official Review · Reviewer_mVL3 · 2025-11-03

**Soundness:** 2
**Presentation:** 2
**Contribution:** 2
**Rating:** 2
**Confidence:** 5

**Summary:**

LiDAR 3D detectors are essential for autonomous driving, and missed detections pose severe safety risks. Existing adversarial point attacks seldom remove objects or translate to the physical world. This paper presents Phy3DAdvGen, a text-to-3D adversarial method that generates physically realizable pedestrian models invisible to LiDAR. It optimizes text prompts and restricts outputs to a pool of thirteen real objects for physical feasibility. In CARLA, the authors manipulate topology, connectivity, intensity, and object combinations to study vulnerabilities. Phy3DAdvGen iteratively refines verbs, objects, and poses to produce LiDAR-invisible pedestrians. Experiments show these models evade six state-of-the-art LiDAR detectors in simulation and physical tests.

**Strengths:**

(1) This idea is straightforward and can be easily understood.

(2) The experiments consider multiple LiDAR 3D detectors, with seemingly more rigorous evaluation methodologies than many previous papers in this area.

**Weaknesses:**

(1) My major concern about this paper is the novelty issues. The authors claim that their main contribution lies in the ability to use prompts to generate 3D adversarial examples. However, the entire text-to-3D adversarial generation method is largely built upon existing work, particularly Gaussian Splatting (e.g., LGM, Tang et al., 2024) for generating malicious examples. Therefore, the novelty and contribution of this work are quite limited, as it primarily extends existing methods without introducing substantial new techniques or insights.

(2) The threat model is unclear. For example, is this a black-box or white-box attack? How much knowledge does the attacker have about the victim models (i.e., LiDAR detectors) and the associated datasets? If this is a white-box attack, how could it realistically be deployed in a real-world scenario? I recommend that the authors include a dedicated subsection titled “Threat Model” to clearly define their assumptions regarding the attacker’s knowledge, capabilities, and goals. This addition would help readers better understand the scope and practicality of the proposed method.

(3) The experiments are not sufficient. For example, the paper only briefly discusses transferability in Table 4, where the evaluation is conducted on six LiDAR detectors on average. More thorough experiments are needed to analyze how adversarial examples generated on one model can be transferred and applied to other models. A detailed discussion of cross-model and cross-dataset transferability would significantly strengthen the experimental section and the overall credibility of the proposed approach.

(4) Finally, the writing quality requires significant improvement. In several critical parts of the paper, it is difficult to understand what the authors actually did in terms of experimentation and analysis, as well as what motivated their design and methodological choices.

**Questions:**

Please refer to my comments for more details.

---

> ### Author Response · Authors · 2025-11-26
> **Response to Reviewer mVL3 (1/4)**
>
> **W1: Novelty concern**
>
> **A1**:  Thank you for the thoughtful comment and for highlighting this concern. We acknowledge that our framework leverages existing text-to-3D generation backbones (e.g., LGM). However, the core contribution of this work does not lie in proposing a new generative architecture, but in introducing a **new adversarial paradigm for LiDAR-based 3D detector**—i.e., generating *semantically controlled adversarial 3D objects* through prompt optimization, rather than perturbing existing point clouds or meshes. This direction is inspired by recent findings showing that text-to-image diffusion models exhibit inherent adversarial potential against 2D detectors [1].
>
> We believe this work serves as an **initial exploration of an emerging 3D adversarial attack paradigm**, supported by:
>
> - a systematic empirical analysis in CARLA identifying **vulnerability factors** (e.g., topology, Connectivity , intensity, object combinations) that can trigger real-world detection failures;
> - adversarial optimization in a **discrete semantic prompt space (verb–object–pose)**, rather than continuous geometric perturbations;
> - **text-conditioned 3D adversarial object generation**, instead of modifying captured sensor data;
> - **digital to physical transferability without 3D printing**, achieved through prompt-aligned assembly of real-world objects.
>
> These capabilities are fundamentally different from prior point-based, texture-based, or geometry-modification attacks and cannot be easily achieved by simply extending perturbation-based approaches. We believe this work meaningfully expands the threat model for autonomous driving systems and introduces a new class of generative physical attacks with practical implications.
>
> [1]. Sato T, Yue J, Chen N, et al. Intriguing properties of diffusion models: An empirical study of the natural attack capability in text-to-image generative models. In *CVPR*, 2024.

---

> > ### Author Response · Authors · 2025-11-26
> > **Response to Reviewer mVL3 (2/4)**
> >
> > **W2: Threat model clarity**
> >
> > **A2**: We thank the reviewer for raising this important point. our method operates in a **white-box generation stage followed by black-box physical deployment**, meaning the attacker only requires model access *during optimization*, but no access is assumed when performing the real-world attack. We will add the following subsection to the revised manuscript to clearly state the assumptions, attacker knowledge, and deployment constraints.
> >
> > #### **Threat Model**
> >
> > Our framework adopts a **white-box to black-box transfer attack setting**, consistent with prior physical adversarial studies. During the generation stage, the attacker is assumed to have access to a *surrogate* LiDAR detector, enabling gradient-based optimization over the semantic prompt space (verb–object–pose) to guide the text-to-3D generator toward adversarial geometry. After optimization, the attack transitions to a **fully black-box deployment**: the attacker has no access to the victim model’s architecture, parameters, outputs, or training data. The resulting adversarial object can be physically instantiated either via 3D printing or by assembling accessible real-world items that reproduce the optimized configuration (e.g., an umbrella placed above the head). This design ensures that the attack remains **fully passive at deployment**, requiring no interaction with the perception system and aligning with realistic adversarial capabilities and real-world deployment constraints.

---

> ### Author Response · Authors · 2025-11-26
> **Response to Reviewer mVL3 (3/4)**
>
> **W3: Transferability experiments**
>
> **A3**: We appreciate the reviewer’s insightful suggestion regarding expanding the **transferability evaluation**. We agree that transferability analysis is important for understanding generalization. Our current evaluation includes a **cross-model benchmark** covering six representative LiDAR-based detection architectures (Table 4). For completeness, we include the expanded Table 4 results below and will provide the full version in the **supplementary material**. In addition to Table 4, **Table 2 provides another perspective on transferability**, showing that a **single optimized adversarial object** remains effective across varying **LiDAR resolutions**, **motion patterns**, and **environmental conditions**. This experiment also represents **deployment-level transferability**, which is highly relevant in **physical-world attack scenarios**.
>
> Following the reviewer’s suggestion, we have also added **cross-dataset transferability experiments**, where adversarial objects optimized in **CARLA** are directly rendered into **KITTI** and **nuScenes**. As shown in the Table below, the attack remains highly effective across datasets, with higher success rates observed on **nuScenes**—potentially related to its lower-density **32-beam LiDAR configuration**.
>
> The transferability results highlight a key difference between our generative adversarial objects and prior point-level perturbation attacks. Whereas perturbation-based methods often overfit a specific model, our text-to-3D approach generalizes across detectors, datasets, and sensing conditions because the adversarial signal lies in the **semantic-level geometry** rather than localized point noise. The nuScenes subclass statistics provided in our response to **R3 (egro)–Q3** further support this: although thousands of fine-grained pedestrian instances exist, the mAP drops from **0.879 (generic)** to **0.0–0.02**, showing that detectors rely heavily on canonical pedestrian shapes and struggle with geometric variation. Our generated objects exploit this same bias—remaining **plausible** while subtly deviating from standard geometry—resulting in strong cross-model and real-world transferability. These findings suggest that generative 3D attacks represent a **more scalable and deployable physical threat** than point-level perturbations and expose a vulnerability rooted in LiDAR perception assumptions rather than individual model design. Further supporting evidence regarding transferability is also discussed in our response to R3 (egro)–Q1, which aligns with and strengthens the conclusion presented here.
>
> |      Prompt Strategy      | PointRCNN† |  IA-SSD  | SAFDNet  | VoxelNeXt |  HEDNet  |   PDV    |      Mean      |
> | :-----------------------: | :--------: | :------: | :------: | :-------: | :------: | :------: | :------------: |
> |       Random Prompt       |    75.1    |   73.3   |   69.0   |   71.5    |   67.6   |   69.7   |   71.0 ± 6.6   |
> |       LatentPerturb       |    81.7    |   80.7   |   71.3   |   78.1    |   77.3   |   78.1   |  78.0 ± 11.2   |
> |  Phy3DAdvGen (verb-only)  |    66.7    |   64.0   |   61.3   |   60.0    |   62.6   |   61.7   |   62.8 ± 4.7   |
> | Phy3DAdvGen (verb+object) |    82.1    |   74.0   |   80.4   |   75.4    |   76.7   |   75.0   |   77.3 ± 8.8   |
> |      **Phy3DAdvGen**      |  **92.1**  | **87.5** | **86.8** | **82.7**  | **87.9** | **89.1** | **87.7 ± 7.9** |
>
> | Dataset Transfer  | PointRCNN† | IA-SSD | SAFDNet | VoxelNeXt | HEDNet | PDV  |
> | :---------------: | :--------: | :----: | :-----: | :-------: | :----: | :--: |
> |  CARLA to KITTI   |    91.7    |  87.2  |  86.1   |   83.5    |  88.4  | 89.0 |
> | CARLA to nuScenes |    94.0    |  90.2  |  85.1   |   87.8    |  90.4  | 91.6 |

---

> ### Author Response · Authors · 2025-11-26
> **Response to Reviewer mVL3 (4/4)**
>
> **W4: Writing clarity**
>
> **A4**: Thank you for the constructive feedback. We acknowledge that parts of the original manuscript may have been difficult to follow due to the novelty of the proposed attack paradigm and the multi-stage experimental workflow. To improve clarity, we have revised the manuscript to make the methodology and experimental pipeline more explicit and easier to follow. In particular, we have added a dedicated **Threat Model** subsection to better contextualize our design assumptions and evaluation settings. We have also expanded the discussion on transferability. A concise analysis of **cross-model and cross-dataset transferability** has been added to the main manuscript, and the experiments and discussion raised in **R1(mVL3)-Q3** and **R3(egro)-Q4** have been incorporated to ensure these findings are clearly represented in the core narrative. We believe these revisions substantially improve the clarity, readability, and transparency of the paper and help readers more easily understand the experimental design, reasoning process, and methodological choices underlying our approach.

---

### Author Response · Authors · 2025-12-04
**Rebuttal Final summary**

**Dear Area Chair**,

Thank you for your time and efforts throughout the review process. We sincerely appreciate all reviewers’ feedback. Across all three reviews, the novelty, clarity, and empirical rigor of our work are consistently acknowledged:

- **Novel paradigm**: Phy3DAdvGen is identified as the **first text-to-3D, prompt-driven, physically realizable adversarial attack** on LiDAR, introducing a new attack class that goes beyond point-cloud perturbations by **generating new adversarial objects**. While one reviewer initially questioned novelty, our rebuttal clarifies that prior 3D attacks modify existing scenes, whereas Phy3DAdvGen is the first to **generate and insert adversarial objects via text-to-3D**, a point explicitly affirmed by the other reviewers.
- **Physical realism & effectiveness**: Reviewers agree that our generated human–object compositions are **physically feasible**, successfully making pedestrians vanish from **six SOTA LiDAR detectors** in both simulation and real-world tests.
- **Rigorous evaluation**: The paper is consistently commended for its **broad and systematic experiments**, including cross-detector evaluations, real-world deployment, cross-dataset transfer (CARLA → KITTI/nuScenes), and extensive ablations.
- **Clarity**: Reviewers describe the method as **straightforward, well-structured, and easy to understand**, with clear motivation and algorithmic flow.

We also fully addressed all reviewer concerns in the rebuttal:

- **Simulation–reality gap**: We added real-world LiDAR scans and analysis demonstrating that the semantic-level geometric deviations responsible for detector failure persist beyond simulation. We also clarified that our insertion and evaluation protocols follow established practices such as GT-Paste.
- **Dependence on a specific generator**: We explained that Phy3DAdvGen is **generator-agnostic**. Optimization operates entirely in the **discrete semantic prompt space** (verb–object–pose), not in model-dependent latent space. Additional tests with varied prompts and object configurations show consistent evasion, confirming that the effect stems from semantic geometry rather than LGM-specific artifacts.
- **Threat model and clarity**: We added a dedicated threat-model section describing the **white-box generation → black-box deployment** setting, and reorganized the methodology and transferability sections, **which considerably improves clarity**.

Overall, the paper introduces a **new and impactful generative attack paradigm** for autonomous driving—moving from perturbing existing scenes to **generating physically realizable adversarial objects**. Combined with extensive experiments and rebuttal clarifications, our work provides a timely and rigorous contribution that addresses all reviewer concerns, **including initial questions about novelty**.

---

### Note · Program_Chairs · 2026-01-17
**Submission Desk Rejected by Program Chairs**

The following references in this submission do not refer to real documents and/or have major errors in bibliographic information:

 Chen He, Zhaoxiang Yang, Yabin Zhang, Le Sun, and Liang Wang. Pdv: A flexible and effective 3d object detection framework for autonomous driving. In ECCV, 2022.